# Neutrophil-to-Lymphocyte Ratio and Systemic Immune-Inflammation Index—Biomarkers in Interstitial Lung Disease

**DOI:** 10.3390/medicina56080381

**Published:** 2020-07-29

**Authors:** Victoria Maria Ruta, Adina Milena Man, Teodora Gabriela Alexescu, Nicoleta Stefania Motoc, Simina Tarmure, Rodica Ana Ungur, Doina Adina Todea, Sorina Cezara Coste, Dan Valean, Monica Carmen Pop

**Affiliations:** 1Faculty of Medicine, “Iuliu Hatieganu” University of Medicine and Pharmacy, 400000 Cluj Napoca, Romania; victoria.suteu@yahoo.com; 2Department of Pneumology, “Iuliu Hatieganu” University of Medicine and Pharmacy, 400000 Cluj Napoca, Romania; manmilena50@yahoo.com (A.M.M.); motoc_nicoleta@yahoo.com (N.S.M.); doina_adina@yahoo.com (D.A.T.); cpop@umfcluj.ro (M.C.P.); 3Department of Internal Medicine, “Iuliu Hatieganu” University of Medicine and Pharmacy, 400000 Cluj Napoca, Romania; siminatarmure@yahoo.com (S.T.); sorina.cezara@gmail.com (S.C.C.); 4Department of Medical Rehabilitation, “Iuliu Hatieganu” University of Medicine and Pharmacy, 400000 Cluj Napoca, Romania; ungurmed@yahoo.com; 5Department of General Surgery, “Iuliu Hatieganu” University of Medicine and Pharmacy, 400000 Cluj Napoca, Romania; valean.d92@gmail.com

**Keywords:** neutrophil-to-lymphocyte ratio (NLR), systemic immune-inflammation index SII 2, idiopathic pulmonary fibrosis (IPF), interstitial lung disease (ILD), connective tissue diseases (CTD), usual interstitial pneumonia (UIP), non-specific interstitial pneumonia (NSIP)

## Abstract

*Background and objectives:* The aims of the study were to evaluate the utility of neutrophil-to-lymphocyte ratio (NLR) and the systemic immune-inflammation index (SII) as inflammation markers and prognostic factors in patients with known interstitial lung disease secondary to connective tissue diseases (CTD-ILD) compared with idiopathic pulmonary fibrosis (IPF). *Materials and Methods:* Forty-two patients with known interstitial lung disease (21 with IPF and 21 with CTD-ILD) and 42 control matched healthy patients were included. The NLR was calculated as the absolute neutrophil count divided by the absolute lymphocyte count, and the SII was calculated as follows: SII = platelets × neutrophils/lymphocytes, with the data being obtained from the patients data charts at admission, before any treatment. *Results:* our hypothesis was that in patients with interstitial lung disease NLR and SII would have higher values compared with patients with CTD-ILD or control healthy patients. The mean NLR value was 3.01 (±1.35) among patients with idiopathic pulmonary fibrosis, and 2.38 (±1.08) among patients with CTD-ILD without significant statistical difference (*p* = 0.92). There was however a clinically significant statistical difference when compared with the control group, where NLR was 2.00 (±1.05) (*p* = 0.003). SII values were 619.37 (±329.51) in patients with IPF, 671.55 (±365.73) in CTD-ILD group and 569.73 (±326.67) in healthy subjects (*p* = 0.13) *Conclusions:* A mean NLR value of 2.8 and a SII value over 500 in patients with connective diseases can become a marker of pulmonary interstitial involvement. In the context of non-exacerbated interstitial lung disease, NLR and SII have reduced numerical values, without being statistically correlated with prognosis when we compared with patients with connective tissue diseases without exacerbation or with healthy people, the cut off being of 2.4. However subsequent studies in larger patient samples might provide changes in these cut-off values.

## 1. Introduction

The terms “interstitial lung disease” (ILD) or “diffuse parenchymal lung disease” are a large and heterogeneous group of diseases that affect the interstitium [1,2]. The prevalence of ILD has been estimated as 81/100,000 in males and 67/100,000 in females [2]. These diseases have different clinical, biological, and imagistic features that could help in their differentiation and the final diagnosis. High-resolution computed tomography (HRCT) is a valuable tool for the diagnosis and monitoring of ILD [1,3]. The traits on the HRCT (reticulation, traction bronchiectasis, ground glass or honey combing) in different combinations form a certain CT pattern that could be differentiated from several clinical entities or diseases. For instance, a specific form of chronic fibrosing interstitial pneumonia, of unknown etiology, limited to the lung is a usual interstitial pneumonia pattern (UIP). The most frequent clinical disease that has a UIP pattern on HRCT scanner is idiopathic pulmonary fibrosis (IPF) [4]. IPF prevalence ranges from 1.25 to 23.4 cases per 100,000 population. It affects adults, especially men, aged over 50 years and has a poor prognosis, as death typically occurs at 2–4 years after diagnosis [5]. It is influenced by personal factors (smoking status, comorbidities, etc.). Up to 10% of patients with a CT pattern of UIP subsequently develop systemic diseases [6]. Connective tissue disease (CTD) is a group of autoimmune disorders from which the most common, rheumatoid arthritis (RA), affects between 0.5% to 2% of the general population in the USA [3,6]. ILD occurs in 15% of patients with connective disease and are called connective tissue disease associated interstitial lung disease (CTD-ILD) [3]. The HRCT changes that are found in CTD-ILD form a pattern called “non-specific interstitial pneumonia” (NSIP). An association with polymyositis/dermatomyositis, systemic sclerosis and Sjogren syndrome is often observed, which should facilitate diagnosis. Immune/systemic inflammatory response injuries and fibrosis [3,7] can lead to a disturbance of the normal lung architecture described as HRCT [8]. The lung ultrasound (LUS) is a non-invasive imaging technique that may become an important instrument to detect patients with suspected ILD [8]. The multidisciplinary discussions (MDD) could suggest diagnosis with or without the need for invasive investigations (e.g., lung biopsy). Accurate diagnosis is extremely important due to the therapeutic options and the long-term prognosis [9].

There is an increasing interest in research for a better understanding of ILD to identify prognostic and predictive factors with cheap, simple and easier to obtain tests such as blood tests. A number of studies described in ILD mild hematological changes, such as anemia, leukocytosis, leukopenia, lymphopenia, eosinophilia, thrombocytopenia.

**The neutrophil-to-lymphocyte ratio (NLR)** is a marker of subclinical inflammation, with a higher values in exacerbations, in many lung diseases such as cancer, COPD, asthma, bronchiectasis, obstructive sleep apnea severity, obesity, and many cardiac diseases (ischemic heart disease, myocardial infraction) [10,11,12]. Patients with a higher NLR value have a relatively low lymphocyte count and a high neutrophil count and indirectly evaluates inflammatory status as well as cell-mediated immunity [13].

**The systemic immune-inflammation (SII) index**, as a novel inflammation-related index, is a comprehensive combination based on peripheral lymphocyte, neutrophil, and platelet counts. It is calculated as follows: SII = platelet count × neutrophil/lymphocyte count [14]. According to previous studies, SII may have a high prognostic value in cancer patients, and an elevated pretreatment SII is associated with poor outcomes in cancer patients [15].

Unlike other inflammatory biomarkers, such as erythrocyte sedimentation rate (ESR) and C-reactive protein (CRP), the NLR and the systemic immune-inflammation index are derived from complete routine tests. They require no special technique, making them therefore rapidly accessible [16]. Our hypothesis was that, as in idiopathic pulmonary fibrosis the inflammation is low, and fibrosis is predominantly a pathogenetic finding localized in the lung, NLR should have smaller values than in CTD-ILD where the inflammation is systemic.

The aims of the study are to evaluate the value and the cut-off of the neutrophil-to-lymphocyte ratio (NRL) and of the systemic immune-inflammation index (SII) as inflammatory status markers in patients with IPF compared with known interstitial lung diseases (CTD-ILD) and healthy people. Their possible role as a prognostic factor was also briefly analyzed.

## 2. Materials and Methods

This is a retrospective observational study conducted in the Tertiary Teaching Pulmonology Hospital from Cluj-Napoca over a period of 18 months, from January 2018 until June 2019. All patients with known interstitial lung disease (diagnosed made according to the current existing guidelines), over 18 years old, admitted consecutively in the pulmonology wards were included. Considering the CT pattern, the patients were divided into two groups: One group with confirmed typical UIP pattern (diagnosed with idiopathic pulmonary fibrosis in MDD) and the second group with NSIP pattern (CTD-ILD). All patients with other chronic respiratory diseases (e.g., asthma, COPD), including patients with sleep apnea and obesity hypoventilation syndrome, patients with documented clinical allergies, neoplastic diseases, hematological disorders, and other acute diseases (pneumonia, hypersensitive pneumonia or pulmonary embolism) that could have explained the inflammation were excluded. A control healthy group matched for age and sex was also evaluated for comparation. All patients signed an informed consent. The study was approved by the Ethics Committee of the “Iuliu Hațieganu” University of Medicine and Pharmacy, no. (2807/28.06.2017), and was reapproved by the Ethics Committee of the Hospital no. 4/2020.

Study protocol: Demographic and laboratory data were collected from all charts of patients who were included in the study. Blood samples were collected from all the patients at admission to the hospital. Complete blood cell counts, and differential values were recorded. The NLR ratio was defined as the absolute count of neutrophils divided by the absolute count of lymphocytes. The systemic immune-inflammation index was calculated according to the existing formula: SII = platelets × neutrophils/lymphocytes. The absolute neutrophil count cut-off was established according to the cut-offs accepted in Romania, between: (1.8–7.3/* 10^3^/uL) neutrophils, lymphocytes (1.5–4/* 10^3^/uL), thrombocytes (140–440/* 10^3^/uL), ESR (6–11 mm/1 h), CRP (0–6 mg/L) and LDH (230–460/U/L). HRCT pattern was recorded in standard of care, and multidisciplinary discussions (MDD) made the final diagnosis according with international guidelines. The NLR and SII cut-off for healthy subjects were 2.00 (±1.05) and 569.73 (±326.67), respectively. The data were collected, and the database was created using Microsoft Excel 2017 software.

Statistical analysis: Statistical analysis was performed using IBM SPSS v20.0 (Armonk, NY, USA) for Windows. The normal distribution of the data was checked with the Shapiro–Wilk test. To evidence the mean value of the parameters, standard deviation was used. The comparison of the mean values between two samples was performed using the Student’s *T*-test, and the ANOVA test was used to compare the mean values between three samples. For the comparison of two continuous quantitative variables, the Spearman–Rho test was used. Quantitative data was described using mean ± standard deviation, minimum, maximum, median and the interval between 25th percentile (Q1) and 75th percentile (Q3). Qualitative data was described using counts and percentages; the link between two qualitative data was tested using the Chi2 test with a significance level of 0.05.

Results: The characteristics of all the included patients (42 patients with interstitial lung disease—group I and 42 healthy controls—group II) are shown in Table 1. The age and sex matched controls were included. Characteristics of the studied groups are shown in Table 1. In the first group there were 24 (57.14%) patients with IPF and 18 patients (42.86%) with CTD-ILD. There was a statistically significant difference in age between the two ILD patients groups: in the IPF group the mean age was 65.79 years, with a minimum of 50 and a maximum of 83 years old, while in the CTD-ILD group mean age was 57.61 years, the youngest patient being aged 31 years, and the oldest patient 79 years.

At a subgroup analysis IPF versus CTD-ILD patients there was not any clinical statistical difference in the two markers NLR and SII (see Table 2). During hospitalization 6 deaths were recorded in the IPF group. Mean NLR value was 3.01 (±1.35) in the patients that survived and 2.32 (±0.67) in the patients that died. There was, however, a significantly statistical difference in both NLR and lymphocytes values in patients with ILD when compared with healthy subjects (Table 1). In the control group mean NLR was 2.00 (±1.05), having lower values than in patients with both IPF and CTD-ILD (*p* = 0.003). Although not significant when comparing smoking status among patients with interstitial lung diseases versus healthy subjects, it is clinically significant at a subgroup analysis (see Table 2 and Table 3).

When looking at the SII there was not a statistically significant difference in the analyzed groups: in the IPF group the mean value was 645.88 (±354.55) among patients that survived and was 539.83 (±248.88) in the deceased patients, while in patients with CTD-ILD it was 671.55 (±365.73) (*p* = 0.41). There was no statistical difference between patients with ILD and healthy subjects (*p* = 0.13) (see Table 4).

## 3. Discussion

The present study aimed to analyze the utility of NLR and SII as markers of inflammation and prognostic factors in patients with known interstitial disease (IPF and CTD-ILD) versus control groups. These rapid and easy-to-use biomarkers are minimally influenced by physiological, pathological and physical factors [17] and could evaluate severity and immediate mortality risk. We did not find any statistically significant difference in NLR and SII in the analyzed groups, nor could we associate a higher value with a poor outcome. The mean NLR value in both groups was 2.82. As IPF is a more “localized disease”, with more fibrosis and less inflammation, we would have expected “near” normal values, while in patients with CTD-ILD and in those with poor outcome (death) we would have expected to have higher values for NRL and SII.

There was not a statistically significant difference in SII between the analyzed group and healthy subjects, suggesting that this variable could have less utility as a marker for the existing subclinical inflammation in interstitial lung disease. Neither NLR nor SII correlated with classical inflammation markers such as C-reactive protein and erythrocyte sedimentation rate.

As expected, age was statistically different between the two groups as patients with IPF are much older and typically male compared with CTD-ILD. Smoking was also more frequent in the IPF group, as the patients tend to be heavy smokers (*p* = 0.02).

Inflammation probably plays a role in the pathogenesis and progression of pulmonary fibrosis. However, the mechanisms related to IPF and non-specific pulmonary fibrosis are not well characterized and differentiated. Nevertheless, recent data suggest that IPF pathophysiology is rather a product of fibroblast dysfunction than persistent inflammation. Inflammation in the pathogenesis of the disease is attributed to atypical patients and to the biological samples obtained during disease exacerbations [17].

Patho-physiologically, inflammation induces an increase in neutrophil and thrombocyte count, accompanied by a decrease in lymphocyte count [14]. Neutrophils play a more important role in inflammatory disorders than macrophages; they are the distinctive sign of acute inflammation [18]. Neutrophils are immature phagocytes, which release proteolytic enzymes and free radicals that enhance the inflammatory processes, and their persistence causes anarchic structural changes with unpredictable local lesions [19].

The literature reports different NLR values, using different methods, in various populations (neoplastic or not). Currently, there is no universal value available. The accepted normal NLR values in an adult, non-geriatric population in good health vary between 0.78 and 3.53 [20]. The group of patients selected in our study had a mean age of 65.79 (±7.64) for UIP in IPF and 57.61 (±13.16) for NSIP in CTD-ILD, fitting within the limits accepted by the literature as “normality”.

On the other hand, this fit within the limits might be deceiving. Studies conducted in patients with liver fibrosis, which evaluated the NLR as a marker for the severity of liver fibrosis in patients with chronic viral hepatitis B (CHB), showed that CHB patients with advanced fibrosis have a significantly lower NLR than CHB patients without minimal fibrosis [18]. This might explain the lower mean NLR in UIP patients dying in a relatively short time, and subsequently, it would also be interesting to evaluate the extension of pulmonary fibrosis based on the HRCT score and NLR ratio. In addition, the lower NLR value in deceased patients supports the presence of a non-inflammatory component in the disease progression. Patients with a predominantly fibrosing reaction have a much worse prognosis compared to patients with a predominant inflammatory reaction [21]. The cut-off value is highly controversial; for example, in patients with systemic involvement such as systemic sclerosis (SSc), NLR values higher than 2.59 were useful in the prediction of interstitial lung disease (ILD). ILD is one of the most common and important complications of SSc. Moderate-to-severe pulmonary fibrosis is found in approximately 25% of SSc cases. These patients may remain asymptomatic [20]. Non-Specific interstitial pneumonia (NSIP) is the most frequent ILD pattern in connective tissue diseases (CTDs) with lung involvement, but all patterns can be observed in different CTD [22]. All patients with systemic involvement in our study had NLR values over 2.5. Increased NLR values can be used as a marker similar to CRP in determining extensive inflammation [23] in patients with connective tissue diseases [19] because an increase in CRP may occur late during the evolution of the disease compared to NLR increase [24]. Although there are few data on using NLR and SII as inflammation markers in IPF, our data are similar with those already reported 2.39 ± 1.1 Niksarlıoğlu study [25], 2.39 in Karakut study [26]. Bohrade and colleagues reported that using NLR changes in time could be used to predict the clinical outcomes. In a Phase III pirfenidone trial, outcomes were poorer in IPF patients with greatest increase in NLR or PLR over 12 months vs. the other quartiles [27]. Increased values also correlate with disease severity [26].

Starting from these findings, it would be useful in the future to investigate patients with known systemic involvement, with NLR values over 2.3–2.5, compared with other markers of inflammations and corelated with HRCT findings for possible interstitial involvement, with a higher risk of subsequent complications.

Furthermore, it is known that inflammatory markers increase with population aging, growing levels of obesity and increasing osteoarthritis being part of the aging process and becoming a risk factor for morbidity and mortality among the elderly [28,29]. NLR is a primary marker for this biological process. This theory might explain the slightly higher mean NLR value, although statistically insignificant, in the group with idiopathic fibrosis compared to NSIP [28]. Moreover, scientific evidence supports that stress can activate the inflammatory cascade [29]. Inflammation is a common process, characterized by vasodilatation, leukocyte infiltrate and release of cytokines, which in turn can initiate or maintain the inflammatory process [27]. In addition, the neutrophil, lymphocyte, eosinophil cascade can promote systemic inflammation. This is why NLR, as a rudimentary marker, might represent an indicator of the disease evolution. The assessments performed in this regard, such as NLR at diagnosis ≥5.9, should draw the clinician’s attention because they can predict mortality in IPF [30,31]. Microbiota colonization in the airways acts like a trigger of neutrophil inflammation. Various changes in the microbiome might favor the chemotactic activity of neutrophils, with their increase in the lung [32]. This group of patients will be subsequently evaluated for this correlation as well. In addition, the increase of NLR is also associated with the diagnosis of bacterial infection in exacerbated patients [33]. Low NLR values could be used to guide adequate antibiotic use. We consider that an NLR value ≥5 can be a better marker than CRP, having the capacity to predict the presence of infection, at a lower cost [34].

The systemic immune-inflammation index (SII), a parameter that integrates three types of inflammatory cells (lymphocytes, neutrophils and thrombocytes), has proved to be promising and has been so far evaluated only in the context of neoplasia or the risk for neoplasia depending on different cut-offs [33]. A higher SII usually indicates a stronger inflammatory response and a weaker immune response of patients [35]. It is interpreted, in the context of increased values, as an independent risk factor for the development of cancer. However, a SII ≥330 has been associated with cirrhosis and, implicitly, with liver fibrosis [35,36]. These results suggest that SII might be a more objective marker reflecting the balance between the inflammatory response state and host immunity more accurately than NLR [36]. Another essential benefit that a biomarker could provide is prognostic information, through its change over time, particularly if the information can be rapidly obtained. Routine tests, including HRCT (high-resolution computed tomography) and PFT (pulmonary function test), provide uncertain results regarding the prediction of the outcome in the long term [37,38,39].

Limitations of the study: Involvement of a single Romanian respiratory disease center, the retrospective nature, the 1-year duration of the study, the relatively small number of cases, and the rare pathology studied.

## 4. Conclusions

A mean NLR value of 2.8 and a SII value over 600 in patients with interstitial lung disease without exacerbation could probably be a marker of subclinical inflammation, as it seems to have lower values in healthy subjects. Interestingly, this does not correlate with other inflammatory markers, but it could be used in conjunction with other biomarkers that are more expensive, more sophisticated that and cannot be done routinely in any lab. Subsequent studies in larger patient samples might provide changes in these cut-off values.

## Figures and Tables

**Table 1 medicina-56-00381-t001:** Basic characteristics of the studied groups.

Parameters	Idiopathic Pulmonary Fibrosis (*n* = 24)	Connective Tissue Disease Associated Interstitial Lung Disease (*n* = 18)	Control Group (*n* = 50)	*p* Value
**Age (years)**	65.79 (±7.64)	57.61 (±13.16)	55.04 (±3.27)	**0.001**
**Male**	14 (58.3%)	11 (61.1%)	24 (48%)	0.64
**Female**	10 (41.7%)	7 (38.9%)	26 (52%)	
**Active smoker**	5 (20.8%)	5 (27.8%)	12 (24%)	0.27
**Ex-smoker**	6 (25%)	0 (0%)	5 (10%)	0.13
**Platelets**	221.45 (±60.12)	246.83 (±93.41)	250.98 (±49.36)	0.16
**SII ^1^**	619.37 (±329.51)	671.55 (±365.73)	569.73 (±326.67)	0.13
**Neutrophils**	4.68 (±0.97)	4.56 (±2.34)	4.30 (±1.59)	0.61
**Lymphocytes**	1.85 (±0.59)	1.71 (±0.68)	2.36 (±0.77)	**0.001**
**NLR ^2^**	2.84 (±1.24)	2.80 (±1.08)	2.00 (±1.05)	**0.003**

^1^ SII: systemic immune-inflammation index; ^2^ NLR: neutrophil-to-lymphocyte ratio.

**Table 2 medicina-56-00381-t002:** Characteristics of idiopathic pulmonary fibrosis (IPF) and connective tissue disease-associated interstitial lung disease (CTD-ILD) patients.

Parametes	Idiopathic Pulmonary Fibrosis (*n* = 24)	Connective Tissue Disease-Associated Interstitial Lung Disease (*n* = 18)	*p* Value
**Age (years)**	65.79 (±7.64)	57.61 (±13.16)	**0.023**
**Male**	14 (58.3%)	11 (61.1%)	0.85
**Female**	10 (41.7%)	7 (38.9%)	
**Active smoker**	5 (20.8%)	5 (27.8%)	0.48
**Ex-smoker**	6 (25%)	0 (0%)	**0.02**
**Platelets**	221.45 (±60.12)	246.83 (±93.41)	0.29
**SII ^1^**	619.37 (±329.51)	671.55 (±365.73)	0.67
**Neutrophis**	4.68 (±0.97)	4.56 (±2.34)	0.82
**WBC ^2^**	1.85 (±0.59)	1.71 (±0.68)	0.48
**NLR ^3^**	2.84 (±1.24)	2.80 (±1.08)	0.92

^1^ SII: systemic immune-inflammation index; ^2^ WBC: white blood cells; ^3^ NLR: neutrophil-to-lymphocyte ratio.

**Table 3 medicina-56-00381-t003:** Interpretation of ^1^ NLR values by groups.

	N	Mean	Std. Deviation	Std. Error	Interval for Mean		Minimum	Maximum
					Lower Bound	Upper Bound		
**IPF ^2^**	18	3.0139	1.35215	0.31871	2.3415	3.6863	1.39	5.98
**CTD-ILD ^3^**	18	2.8067	1.08980	0.25687	2.2647	3.3486	1.57	5.24
**IPF Death**	6	2.3217	0.65755	0.26845	1.6316	3.0117	1.65	3.31
**Total**	42	2.8262	1.16454	0.17969	2.4633	3.1891	1.39	5.98

^1^ NLR: neutrophil-to-lymphocyte ratio; ^2^ IPF: idiopathic pulmonary fibrosis; ^3^ CTD-ILD: connective tissue disease-associated interstitial lung disease.

**Table 4 medicina-56-00381-t004:** Interpretation of mean ^1^ SII values.

	N	Mean	Std. Deviation	Std. Error	Interval for Mean		Minimum	Maximum
					Lower Bound	Upper Bound		
**IPF ^2^**	18	645.8889	354.55322	83.56900	469.5737	822.2041	300.00	1643.00
**NSIP ^3^**	18	671.5556	365.73475	86.20451	489.6799	853.4312	351.00	1762.00
**IPF Death**	6	539.8333	248.88826	101.60821	278.6411	801.0255	308.00	1005.00
**Total**	42	641.7381	342.13591	52.79727	535.1210	748.3551	300.00	1762.00

^1^ SII: systemic immune-inflammation index; ^2^ IPF: idiopathic pulmonary fibrosis; ^3^ NSIP: non-specific interstitial pneumonia.

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
