# Peer review of "Neutrophil-to-Lymphocyte Ratio and Systemic Immune-Inflammation Index—Biomarkers in Interstitial Lung Disease"

_medicina, 2020, doi:10.3390/medicina56080381_

Round 1

Reviewer 1 Report

Ms. Ref. No.: medicina-828792 R2

General comments:

The authors have updated the text in the revised version of their manuscript. Thanks for their clear response for each review. They have added the new data of neutrophil-to-lymphocyte ratio (NLR) and the systemic immune-inflammation index (SII) of control group to compare with idiopathic pulmonary fibrosis (IPF) and connective tissue disease-associated interstitial lung disease (CTD-ILD) whether these indices were regarded as the biomarkers. All concerns have been adequately addressed in the manuscript.

Specific comments:

Minor:

#1. “idiopathic pulmonary fibrosis” and “IPF” should be unified into one of them after described the abbreviation once throughout the text.

#2. Page 2, line 60: “usual interstitial pneumonia” should be changed to “UIP” because the abbreviation was already described.

#3. Page 3, line 97: “CDT-ILD” seems to be miswritten and should be changed to “CTD-ILD”.

#4. Page 6, line 204: “66.16” may be changed to “65.79”.

#5. Page 7, line 274: “can no” seems to be miswritten and should be changed to “cannot”.

Author Response

Response to reviewer:

Minor:

#1 . “idiopathic pulmonary fibrosis” and “IPF” should be unified into one of them after described the abbreviation once throughout the text.

       Answer: The term was unified in the text where appropriate (Line 138, Line 189).

#2. Page 2, line 60: “usual interstitial pneumonia” should be changed to “UIP” because the abbreviation was already described.

       Answer: The term was changed into „UIP”(Line 60).

#3. Page 3, line 97: “CDT-ILD” seems to be miswritten and should be changed to “CTD-ILD”.

       Answer: The term was changed (Line 96).

#4. Page 6, line 204: “66.16” may be changed to “65.79”.

        Answer: The number was changed (Line 203).

#5. Page 7, line 274: “can no” seems to be miswritten and should be changed to “cannot”.

        Answer: „Can no”was changed in „cannot” (Line 274).

Reviewer 2 Report

Not the case. 

Author Response

Thank you ! Me and my colleagues appreciate the time and effort that you dedicated to providing feedback on our manuscript. 

Kind Regards,

Teodora Alexescu

This manuscript is a resubmission of an earlier submission. The following is a list of the peer review reports and author responses from that submission.

Round 1

Reviewer 1 Report

Ms. Ref. No.: medicina-828792

General comments:

The authors evaluated the neutrophil-to-lymphocyte ratio (NLR) and the systemic immune-inflammation index (SII) as the biomarkers to predict interstitial lung disease (ILD), and concluded that both NLR value of 2.8 and SII value over 500 may become the markers. However, the conclusion cannot be drawn and endorsed because there was devoid of comparison of the marker values between patients manifested with and without of ILD in the manuscript. Although the review for each section has not been adequately addressed, several changes are required to update the manuscript.

Specific comments:

Major:

#1. In order to recognize NLR and SII as the biomarkers, both values of NLR and SII in control patients without ILD should be demonstrated and compared with the current data in the text. The results should be analyzed adequately and statistically. The statistical data may help the discussion.

#2. Was there any correlation between NLR, SII and clinical indices of ILD? such as fibrotic scores of computed tomography findings, restrictive disorder of pulmonary function test, and serum markers including Krebs von den Lungen-6 and surfactant protein D. If some correlation was existent, a possibility that NLR and SII become the biomarkers may be raised.

#3. Materials and Methods: Was the data extracted only from patients with idiopathic interstitial pneumonia (IIP)? The authors should explain and describe that clearly in the text. If the authors extracted the data only from patients with IIP, what did the “secondary pulmonary fibrosis” (Page 1 line 24, and Page 4 lines 143 and 145) mean? Did those mean non-IIP?

#4. Materials and Methods: Hematological disease should be added to the “Exclusion criteria”, because it possesses abnormal composition of blood cells.

Minor:

#1. Page 1 line 24, and Page 4 lines 143 and 145: “secondary pulmonary fibrosis” should be reconsidered to describe in the text as mentioned above.

#2. Page 3, line 121: “T test” should be changed to “T-test”.

#4. Page 5, line 169: “66.16” may be changed to “65.79”.

#5. Several abbreviations may be unified when those were once explained, such as UIP rather than IPF (if the data was extracted only from patients with IIP), and NSIP throughout the text.

#6. Page 3, line 116: “idiopathic fibrosis” may be changed to “UIP” as mentioned above.

Reviewer 2 Report

In order to improve the manuscript, I consider that the following minor comments should be answered:

General comments:

  1. The meaning of certain passages is difficult to understand, like aim of the study, so it would be of interest if a revision of the text will be done. I suggest to be used shorter but comprehensive sentences in terms of content.
  2. Regarding the statistical analysis, although it is presented that the Spearman-Rho test was used, the way of presenting the results does not reveal where.
  3. Review the bibliography in terms of accuracy, as some bibliographic citations appear incomplete (eg citation no. 11), some links to online citations appear to be non-functional (eg citation no. 10), and others seem incorrectly assigned in the text (eg citation no. 17).

Specific comments:

  1. In table 1 miss the description of the way in which the values and units of measurement for each parameter are presented.

  1. According to paragraph 102-104 absolute values of neutrophil was established according to lymphocytes, thrombocytes, ESR, CRP and LDH which seems non-sense.

  1. I do not think that the way of presenting the results in table 2 and 3 is the most appropriate for a journal article, many of the values derived from the statistical analysis of the data, which are presented having no relevance for the interpretation of the results. A common table could be made in which the value of the statistical significance when comparing the lots should be presented.